# Diversity of Binucleate *Rhizoctonia* spp. and Population Structure of AG-A in Serbia

**DOI:** 10.3390/jof11060410

**Published:** 2025-05-26

**Authors:** Mira Vojvodić, Brankica Pešić, Petar Mitrović, Ana Marjanović Jeromela, Ivana Vico, Aleksandra Bulajić

**Affiliations:** 1Faculty of Agriculture, University of Belgrade, Nemanjina 6, 11080 Belgrade, Serbia; miravojvodic2510@gmail.com (M.V.); vico@agrif.bg.ac.rs (I.V.); 2Institute of Pesticides and Environmental Protection, Banatska 31b, 11080 Belgrade, Serbia; brankica.pesic@pesting.org.rs; 3Institute of Field and Vegetable Crops, Maksima Gorkog 30, 21000 Novi Sad, Serbia; petar.mitrovic@ifvcns.ns.ac.rs (P.M.); ana.jeromela@ifvcns.ns.ac.rs (A.M.J.)

**Keywords:** binucleate *Rhizoctonia*, haplotypes, aggressiveness

## Abstract

From 2013 to 2021, 112 binucleate *Rhizoctonia* spp. (BNR) isolates were obtained from the strawberry, tomato, pepper, bean, apple, cherry, meadow grass, and soil previously cropped with strawberries from 16 locations in Serbia. Morphological and molecular analyses (ITS, LSU rDNA, *RPB2*, tef-1α, and *atp6*) confirmed infections caused by four BNR AGs: AG-G on the cherry (globally new host), bean, and tomato; AG-U on meadow grass (globally new host) and apple, AG-A on the strawberry (the most frequently isolated), and AG-F on pepper. ITS sequence analysis revealed 24 haplotypes within the worldwide population of BNR AG-A, with Serbian isolates belonging to nine. The aggressiveness of AG-A (ten isolates), AG-G (three isolates), AG-F (one isolate), and AG-U (two isolates) was tested on seedlings of 14 hosts from Poaceae, Brassicaceae, Solanaceae, Asteraceae, Fabaceae, Cucurbitaceae, Apiaceae, and Chenopodiaceae, and on detached leaf petioles of the strawberry, tomato, sunflower, and bean, as well as on two pea cultivars. Sunflower and sugar beet were the most susceptible, with AG-G being the most aggressive and AG-A the least aggressive. AG-A could not infect cabbage, while at least one isolate of each remaining AG infected all tested hosts. The consistency between seedling and petiole tests highlights the latter as a rapid method for evaluating the pathogenicity and aggressiveness of BNR isolates.

## 1. Introduction

*Rhizoctonia* species are soil-borne, mostly pathogenic basidiomycetes that are frequently associated with plant roots. They are very diverse in terms of morphology, host specificity, and pathogenicity and are present in many geographical areas. The host range of *Rhizoctonia* spp. is broad [1] and includes fruit and forest trees, as well as a range of monocotyledonous and dicotyledonous herbaceous plants. The mycelium of *Rhizoctonia* spp. can survive for a long time on plant debris and can be spread with soil particles and sclerotia [2]. The control of diseases caused by *Rhizoctonia* spp. is complex and costly due to their soil-borne nature and long persistence, while the lack of resistant plant varieties and an insufficient number of effective fungicides with expensive applications contribute to this problem [3].

Widespread infections with *Rhizoctonia* spp. are associated with symptoms of stem canker, black scurf, seedling damping-off, foliar blight, leaf spots, and sheath blight in a large number of different plants [1]. Infections can occur at different stages of plant development [3], while roots and stolons can be infected at any time during the growing season. Numerous reports have linked *Rhizoctonia* diseases to significant yield reductions, including up to 30% losses in potatoes [4,5], 30% in soybeans [6], 20% in rice [7], and 40% in oilseed rape [8].

The most important historical principle and criterion for the classification of *Rhizoctonia* spp. is the status of the nuclei, which can be divided into three subgroups: uninucleate, binucleate, and multinucleate *Rhizoctonia* (UNR, BNR, and MNR) [9,10,11]. Each subgroup consists of different anastomosis groups (AGs), which are further subdivided into intraspecific groups (IGs) that differ in colony appearance (e.g., pigmentation, zonation, presence of sclerotia and moniloid cells, and growth rate), physiological characteristics, pathogenicity, and/or fungicide sensitivity [3,9]. BNR (teleomorph *Ceratobasidium* spp.) consists of 19 AGs, referred to as AG-A to AG-W [12,13].

The study of *Rhizoctonia* spp. diversity and population structure is of great importance, and the AGs with the most pronounced diversity include MNR AG-1, AG-2, AG-4, and AG-6, and BNR AG-B and AG-D [1,12,14,15]. Most studies focus on identifying which AGs are present in specific crops, including wheat and clover [16], various vegetables [17], potato and tobacco [18], beans [19], and strawberries [20,21] or soil [22,23,24], while data on the population structure of certain AGs are limited. More detailed analyses of a small number of selected AGs have shown the existence of diversity. For example, haplotype diversity was confirmed in the *R. solani* AG-3 population from the potato, while no genetic diversity was found within tobacco populations [25]. Differences in population structure are reported for *R. solani* AG-1-IA, with 12 and 39 haplotypes identified [26,27]. Haplotype heterogeneity is also detected in *R. cerealis* AG-D [28].

Despite the great diversity of BNRs in terms of their virulence and host range [10], only a few AGs are examined in-depth due to their importance. The BNRs AG-A and AG-G are described as the most common and the most virulent, with the widest host range [12]. In Serbia, data on the diversity of BNRs are limited, and only *Rhizoctonia* spp. AG-A and AG-I have been described as strawberry pathogens [20,29]. Due to the importance of BNR for the production of various crops, the objectives of our study were to (i) isolate *Rhizoctonia* spp. AGs from different diseased host plants and soils of agricultural land in Serbia, (ii) identify isolates based on morphology and nuclear status, (iii) investigate the diversity of morphology within and between different AGs, (iv) test pathogenicity to fulfill Koch’s postulates, (v) perform molecular and phylogenetic characterization based on the sequencing of the ITS (ITS1, 5.8S rDNA, and ITS2) and LSU of the rDNA, *RPB2*, *tef-1α,* and *atp6* genomic regions, (vi) investigate the structure of the BNR AG-A population in Serbia and worldwide based on haplotype analyses, and (vii) compare the host range and aggressiveness of isolates within and between different AGs using different approaches.

## 2. Materials and Methods

### 2.1. Field Surveys, Sample Collection, and Morphological Identification

This survey was conducted during 2013–2021 at 16 locations across Serbia, and symptomatic plant (14 locations) and soil (2 locations) samples were collected in seven different crops: strawberry (*Fragaria x ananassa*), tomato (*Solanum lycopersicum*), pepper (*Capsicum annuum*), bean (*Phaseolus vulgaris*), apple (*Malus domestica*), cherry (*Prunus avium*), and meadow grass (*Poa annua*). At each location, the incidence of the disease was estimated by walking through the crop in a zigzag pattern and randomly rating 100 plants in three replicates. A total of 163 samples of plants with visible symptoms of root and root neck rot, wilting, and decay were collected and analyzed within 48 h. The number of samples at each locality was different and depended on the field size and disease incidence (Table 1). For isolation, small sections (0.5 cm diameter) were taken from the edge of diseased plant tissue, washed in running tap water for 2 h, surface sanitized with 2% sodium hypochlorite for 1 min, plated on potato dextrose agar (PDA, 200 g potato, 20 g dextrose, 17 g agar, and 1 L distilled H_2_O) [30], and incubated at 24 °C for 3 days.

Soil samples from the 10–15 cm depth of the soil profiles, from two fields with strawberry as a previous crop, were collected (approximately 500 g) and transferred to pots and kept in the greenhouse (20 °C). Pathogens were isolated from the soil samples using the bait plant method [30] by sowing (one day after sampling) sterile bean seeds (previously shown to be susceptible to the various BNR AGs). The pots were then irrigated to soil capacity. From symptomatic bait plants in the phenophase of cotyledons, re-isolation was performed using the same method as for isolation.

*Rhizoctonia*-like colonies were selected based on colony appearance and branching pattern. Hyphal tip isolates were obtained after subculturing colonies on water agar (WA, 17 g agar, and 1 L distilled H_2_O) and transferring the isolated tips of individual hyphae to PDA. Selected isolates from each locality were identified based on the number of nuclei in three-day-old hyphal cells, hyphal branching pattern, growth rate, and colony morphology 15 days post-inoculation (dpi) on PDA at 24 °C in the dark [10]. One isolate representing uniform isolates from each site was selected for detailed characterization.

### 2.2. Pathogenicity Testing

Pathogenicity was tested depending on the origin of the isolate. After all tests, the challenging isolates were recovered from symptomatic plants using the same method as for isolation and used for further characterization.

The pathogenicity of 10 selected isolates from the strawberry was tested by inoculating six plants each (replicates) of stolon-propagated disease-free strawberry daughter plants of the cultivar ‘Senga-Sengana’ according to the published method [31]. Negative control plants were inoculated with sterile PDA. Each healthy, non-wounded strawberry plant was transplanted into a pot (12.5 cm diameter and 9 cm height) containing a sterilized commercial substrate, and a 7-day-old *Rhizoctonia* colony (9 cm diameter) was placed in the root zone. Plants were maintained in the greenhouse (23 °C, 12 h photoperiods), and the appearance of symptoms was assessed up to 30 days post-inoculation. Similarly, the pathogenicity of an isolate from meadow grass was tested on five healthy, commercially obtained grass blocks of meadow grass planted in a sterile substrate in pots (15 cm diameter) and grown under greenhouse conditions with regular watering. After two weeks, the grass blocks were transplanted and inoculated with a seven-day-old isolate colony grown on PDA inserted near the roots. The control grass blocks were inoculated with sterile PDA. The appearance of symptoms was monitored at weekly intervals. Re-isolation was performed from all plants with symptoms. The experiment was repeated twice.

The pathogenicity of five isolates from the tomato, pepper, bean, apple, and cherry was tested by inoculating seedlings of the respective host plant. Tomato, pepper, and bean seeds were purchased commercially. Apple and cherry seeds were collected from commercially available ripe fruits and stored in the refrigerator at 5 °C for three months for vernalization and then sawed. Seedlings were inoculated when the first true leaves developed by placing mycelial plugs near the roots during transplanting. Five plugs (2r = 5 mm) from 7-day-old cultures were used per isolate and per seedling, and five seedlings were inoculated with each isolate. Seedlings inoculated with sterile PDA served as the control. Plants were maintained under greenhouse conditions and monitored for symptom development. The experiment was repeated twice.

### 2.3. DNA Amplification and Sequencing

Total genomic DNA was extracted from 100 mg of dry mycelium grown in potato dextrose broth (PDB) for 5 days using the Dneasy Plant Mini Kit (Qiagen, Hilden, Germany) following the manufacturer’s instructions. PCR amplification of the ITS (ITS1, 5.8S rDNA, and ITS2) and LSU of rDNA, *RPB2*, *tef-1α,* and *atp6* used the primers ITS1F (CTTGGTCATTTAGAGGAAGTAA)/ITS4 (TCCTCCGCTTATTGATATGC) [32,33], LROR (GTACCCGCTGAACTTAAGC)/LR5 (ATCCTGAGGGAAACTTC) [34], BRPB26F (TGGGGYATGGNTTGYCCYGC)/BRPB271R (CCCATRGCYTGYTTMCCCAT) [35,36], EF986F (GCYCCYGGHCAYCGTGAYTTYAT)/EF1567R (ACHGTRCCRATACCACCRATCTT) [37], and ATP61 (ATTAATTSWCCWTTAGAWCAATT)/ATP62 (TAATTCTANWGCATCTTTAATRTA) [38], respectively. The conditions were as follows: initial denaturation at 94 °C for 3 min, followed by 35 cycles of denaturation at 95 °C for 30 s, corresponding to annealing temperatures, elongation at 72 °C for 1 min, and final elongation for 10 min at 72 °C. Amplification reactions were performed in a total reaction volume of 25 μL, consisting of 12.5 μL of 2X PCR Master mix (Fermentas, Vilnius, Lithuania), 6.5 μL of RNase-free water, 2.5 μL of both forward and reverse primers (100 pmol/μL, Metabion International, Planegg, Germany), and 1 μL of template DNA. The PCR products were stained with ethidium bromide in a 1% agarose gel for electrophoresis and visualized with a UV transilluminator. The obtained PCR products were sequenced in both directions in an automated sequencer (Automatic Sequencer Macrogen Inc., Maastricht, The Netherlands) using the same primers as for amplification. The consensus sequences were computed using ClustalW [39], integrated into the MEGA X software [40], and deposited in GenBank (http://www.ncbi.nlm.nih.gov) (assessed on 1 May 2022). All generated sequences were compared with each other by calculating nucleotide (nt) identities and with previously deposited isolates of *Rhizoctonia* spp. in GenBank using the similarity search tool BLAST.

### 2.4. Phylogenetic Analyses

The generated ITS sequences of 16 Serbian BNR isolates were analyzed with 21 previously listed type-derived sequences of nine BNRs [12] and one outgroup taxon *Athelia rolfsii* retrieved from GenBank (Table 2). A phylogenetic tree was inferred using the maximum likelihood method implemented in MEGA X software [40]. Distances in the ITS rDNA region were determined using Kimura’s two-parameter model [41], and all sites with gaps were omitted. The reliability of the obtained trees was evaluated using 1000 bootstrap replicates.

### 2.5. Haplotype Analysis of Binucleate Rhizoctonia AG-A Sequences

The analysis of BNR AG-A haplotypes was performed on the basis of all 159 available sequences of the ITS region in GenBank (accessed 30 March 2025), together with 10 Serbian BNR AG-A sequences. Sequences that were too short, those that belonged to the unverified category, or those that had high nucleotide differences and degenerate codons were eliminated, and a total of 76 sequences were included in the analysis. The number of haplotypes (h), haplotype diversity (Hd), number of variable sites (S), and nt diversity (p) of the ITS region were determined using the program DnaSP version 6.0 [46], after which the data on the composition of haplotypes and their frequency were further examined using the PopART software version 1.7. The visualization of haplotypes generated mutual genealogical relationships, using the median-joining network algorithm [47] implemented in the PopART (population analysis with reticulate trees) program [48]. All sequences were compared by calculating nt identities using the MEGA X software [40].

### 2.6. Host Range and Aggressiveness Testing

The host range and aggressiveness of 16 selected Serbian BNR AGs isolates were tested using two methods: inoculation of seedlings (9 isolates) and inoculation of detached leaf petioles (16 isolates).

The inoculation of seedlings of 14 plant species belonging to eight families (wheat (*Triticum aestivum*) and maize (*Zea mays*) from the Poaceae; cabbage (*Brassica oleracea* var. *capitata*) and oilseed rape (*B. napus* var. *oleifera*) from the Brassicaceae, tomato, pepper, and tobacco (*Nicotiana tabacum*) from the Solanaceae, lettuce (*Lactuca sativa*) and sunflower (*Helianthus annuus*) from the Asteraceae, peas (*Pisum sativum*) and beans from the Fabaceae, cucumber (*Cucumis sativus*) from the Cucurbitaceae, carrot (*Daucus carota*) from the Apiaceae, and sugar beet (*Beta vulgaris*) from the Chenopodiaceae) was used to study the host range and aggressiveness of 4 isolates of AG-A, 3 isolates of AG-G, 1 AG-F isolate, and 1 AG-U isolate. Sanitized seeds were incubated on moist filter paper at room temperature for germination until true leaves were formed and inoculated with colony fragments during transplantation, as described for the pathogenicity experiments. The aggressiveness of the isolates was evaluated 7 dpi based on the symptoms observed and according to the scale established for this experiment: 0—no symptoms, 1—up to 30% of roots are necrotic, 2—up to 40% of roots are affected, 3—up to 60% of roots are necrotic, and 4—complete necrosis of roots and decaying of the entire plantlet. Each isolate was inoculated on five seedlings per host plant and the experiment was repeated three times. Seedlings of each host plant inoculated with sterile PDA served as a negative control.

The inoculation of detached leaf petioles of strawberries, tomatoes, sunflowers, beans, and two cultivars of peas, ‘Regina’ and ‘Medicon’, was used to test the aggressiveness of *Rhizoctonia* AG-A (10 isolates), AG-G (3 isolates), AG-F (1 isolate), and AG-U (2 isolates) [49]. Petioles were taken fresh from healthy, greenhouse-grown plants shortened to 20 mm, surface sterilized for 30 s in 50% commercial bleach (2% sodium hypochlorite), and inclined upright in seven-day-old cultures of *Rhizoctonia* isolates grown on PDA at 23 °C in the dark. Eight petioles per isolate were inoculated, while the petioles inclined in the sterile PDA served as negative controls. The inoculated petioles were incubated in a moist chamber and aggressiveness was determined 7 dpi by measuring necrosis length. The experiment was repeated twice.

### 2.7. Statistical Analysis

The colony diameters of the isolates were verified for normality using Colmogorov–Smirnov and Liliefors tests in Graph Pad Software 5.0, Boston, MA, USA, and then analyzed using factorial ANOVA using Statistica 7 (StatSoft, Tulsa, OK, USA). Mean values were compared using Tukey’s test at the significance level of *p* < 0.05.

Ordinal data, as well as necrosis length data in the experiment that failed the normality test, were subjected to the Kruskal–Wallis non-parametric statistical test separately for each host plant. Medians of necrosis length were compared using Dunn’s multiple comparison test at the *p* < 0.05 level of significance. Data were expressed as means ± standard error.

## 3. Results

### 3.1. Symptoms, Morphology, and Pathogenicity

A total of 112 *Rhizoctonia* spp. were obtained from seven host plants (strawberry, tomato, pepper, bean, meadow grass, apple, and cherry) and two soil samples at 16 locations in Serbia (Table 1, Figure 1). Diseased strawberry plants showed leaf necrosis, partial necrosis on the stolons, and root rot, followed by the wilting and decay of the plants, which were often distributed along the rows. Tomatoes, beans, peppers, and meadow grass showed symptoms of leaf chlorosis and necrosis, as well as wilting of the whole plant, with pronounced root rot and necrosis. Three-year-old apple and cherry plants showed aboveground symptoms, such as stunting, leaf chlorosis, and necrotic zones on the leaves, as a consequence of visible necrosis at the stem base, roots, and root hairs (Figure 2A–D). The estimated disease incidence varied among crops, ranging from 5 to 35% (Table 1), with the highest value in strawberry fields (20–35%).

Several *Rhizoctonia*-like isolates forming fast-growing colonies and of uniform appearance and morphology were obtained from each location and host plant, all of which had hyphae with two nuclei (Figure 2I). One representative isolate from each locality and host plant was selected for further study. The isolates were preliminarily identified as one of the four AGs: AG-A, AG-G, AG-F, and AG-U. The isolates of AG-A initially formed white (later beige-colored) colonies (Figure 2E), with no sclerotia. The members of AG-G also initially formed white colonies, which later turned brown (Figure 2F), with moniloid cells (Figure 2J) and sclerotia that became visible after 4 and 7 days, respectively. The members of AG-F initially formed white colonies, which later turned beige (Figure 2G), similar to the members of AG-A, but with moniloid cells and brown sclerotia visible after 4 and 7 days, respectively. The members of AG-U formed initially white colonies, which later turned salmon beige to brown (Figure 2H), with moniloid cells and beige sclerotia visible after 7 and 10 days, respectively. Differences in growth rate between AGs were significant (*p* < 0.01) (Figure 3), with isolates within an AG showing uniform growth rates or minor differences (AG-G). In general, isolates from AG-A grew more slowly than the other three AGs, with isolate 107-13 (AG-A) having the lowest growth rate of 12 mm/day, while isolate 190-18 (AG-F) had the highest growth rate of 22 mm/day.

The pathogenicity of five selected isolates from the tomato, pepper, apple, and cherry was confirmed as they caused visible symptoms of root necrosis on the respective seedlings 7 dpi. Ten isolates from the strawberry caused leaf and root necrosis on strawberry daughter plants at 30 dpi, and one isolate from meadow grass caused leaf and root necrosis on meadow grass at 21 dpi (Figure 2K,L). All developed symptoms resembled natural infection, while the control plants remained symptomless. All isolates were re-isolated from the symptomatic tissue of inoculated plants, thus fulfilling Koch’s postulates.

### 3.2. Molecular and Phylogenetic Characterization

The sequence analyses confirmed the preliminary identification of four Serbian BNR AGs: AG-A, AG-G, AG-F, and AG-U. The BLAST analyses of the ITS sequences of the isolates of AG-A, AG-G, AG-F, and AG-U provided reliable identification as they had the highest nt identities with the respective AG type members and previously identified isolates. Despite numerous attempts and modifications of the protocols, amplification of LSU, *RPB2*, *tef-1α,* and *atp6* was not possible for isolates from all four AGs. The LSU, *tef-1α,* and *RPB2* sequences of Serbian AG-A (Acc. No. MN977412, MT063197, and MT126788) and AG-F (MN977419, MT006340, and MT150071), as well as the LSU and *tef-1α* sequences of AG-G (MN977413 and MT063202) and AG-U (MN977420 and MT063204), showed the highest nt similarities with multiple AGs or with isolates of the unidentified category due to the limited number of available characterized isolates. Amplification of the *atp6* gene was only successful for the AG-U isolate (MT161366), and BLAST analyses showed the highest nt similarity with the only available isolate belonging to AG-U (DQ301578).

The phylogenetic analyses using the maximum likelihood of reference AGs and Serbian BNR isolates resulted in a well-supported phylogenetic tree with a topology consistent with the previous identification of publicly available isolates (Figure 4). All branches of the different AGs confirmed the expected relationships, while the isolates from Serbia were clustered in the AG-A, AG-G, AG-F, and AG-U branches, which were supported with high bootstrap values, thus confirming the previous identification.

### 3.3. Haplotype Structure and Genetic Diversity of BNR AG-A Sequences

A total dataset of 76 ITS sequences of BNR AG-A from Europe, Asia, Africa, Australia, and North and South America, including ten Serbian isolates, showed high genetic diversity and was grouped into 24 haplotypes, with 89 variable positions detected. A nucleotide diversity of 0.01643 (0 to 30 nt difference) and a haplotype diversity of 0.84 indicate a high degree of genetic variation. The most numerous haplotypes consisted of 22 and 20 isolates (Hap_6 and Hap_10) containing isolates from different geographic origins and hosts (Table 3). The Serbian BNR AG-A population of 10 isolates belongs to nine different haplotypes (Hap_1–Hap_9), with a 0 to 5 nt difference and 98.6–99.5% similarity, which are distributed in different growing regions (Figure 1). The haplotype network with 24 haplotypes has a complex network-like structure with several branch points and intermediate nodes (Figure 5). Two major haplotypes, Hap_6 and Hap_16, serve as central nodes, indicating their higher frequency and potential ancestral status within the population. Hap_6 was the most widespread, with 22 sequences from five continents and five hosts, including one Serbian isolate. Hap_6 is connected to multiple other haplotypes, suggesting that it may represent a dominant or ancestral lineage. Hap_10 was the second largest haplotype with 20 isolates, but the isolates originated from a single country and host. The remaining haplotypes comprised a smaller number, mainly single sequences.

### 3.4. Host Range and Aggressiveness of Four Rhizoctonia spp. AGs

The Serbian BNR isolates caused necrosis in the majority of tested host plants 7 dpi, being very aggressive overall and often causing complete necrosis of the roots (Figure 2K–M). Differences in aggressiveness and host range between AGs and even between some of the isolates within AG-A and AG-G as individual AGs have been documented (Figure 6). BNR AG-A showed the narrowest host range and the lowest aggressiveness, while AG-F, AG-G, and AG-U were able to infect all host plants tested. The isolates of AG-G exhibited the highest aggressiveness. The control seedlings showed no symptoms.

Of the four isolates of AG-A, none were able to infect cabbage seedlings, and at least one isolate was unable to infect tobacco, lettuce, pea, cucumber, and maize; however, they were most aggressive on sunflower and sugar beet, and partially on bean seedlings. The three AG-G isolates tested were the most aggressive on the majority of the test plants, with no significant differences among them. The highest disease severity was found in oilseed rape, sunflower, and sugar beet, with an average rate of 3.73, 3.33, and 3.26, respectively. AG-F and AG-U also showed medium to high aggressiveness on all 14 host plants. Statistical analysis of the ordinal data revealed that symptom development was significantly influenced by the host plants (*p* < 0.0001), regardless of the isolate used for inoculations. Sunflower and sugar beet developed the highest disease severity (mean values of 3.12 and 2.6, respectively) (Figure 6), while maize was the least affected (mean value of 0.28). Lettuce and sunflower, as well as peas and beans, showed significant differences in response, although they belong to the same plant family.

With the exception of strawberries, the assessment of aggressiveness, using a test with leaf petioles (Figure 2N), showed similar results. The length of necrosis varied depending on the challenging AG, while no necrosis was visible in control petioles. Isolates of AG-G were found to be the most aggressive on all five host plants (average necrosis length of 13.46 mm), with the exception of AG-A and AG-U on strawberries, where all three AGs showed high and similar aggressiveness (Figure 7). AG-F proved to be the most aggressive on sunflowers (average necrosis length of 13.75 mm) and the least aggressive on strawberries (average necrosis length of 3.01 mm). Statistical analysis of the ordinal data revealed that symptom development was significantly influenced by the host plants (*p* < 0.0001) (Figure 6). The strawberry developed the highest necrosis length (average of 14.73 mm), while beans and peas, as members of the same plant family, showed a similar response (average of 4.10 mm and 4.07 mm, respectively). BNR isolates exhibited a difference in aggressiveness towards the two pea cultivars. The isolates of AG-A and AG-F were more aggressive to ’Regina’, while the isolates of AG-G and AG-U were more aggressive to ’Medicom’.

## 4. Discussion

The study of the diversity of BNRs in Serbia revealed the presence of four AGs, namely AG-A on strawberries, AG-G on tomatoes, beans, and cherries, AG-F on pepper, and AG-U on apples and meadow grass. Before this study, only the presence of AG-A and AG-I, causing black rot, on strawberries was confirmed [20,29], so three additional AGs infecting six host plants are new findings for Serbia. The infection of AG-G, causing root rot in cherries, and AG-U, causing root rot in meadow grass, was detected for the first time in the world. The morphological characteristics of all isolates or all four detected AGs are consistent with those previously reported [50,51,52,53]. The pathogenicity of all isolates was confirmed as they caused necrosis on the non-wounded plants of species from which they were isolated. The identification was also confirmed by BLAST analyses of five molecular markers (ITS, LSU, *RPB2*, *tef-1α*, and *atp6*), as is the case in the study by Gonzalez et al. [15]. Because of the low number of available sequences in GenBank, only ITS sequences were used for phylogenetic studies. A reconstructed phylogenetic tree confirmed the molecular identification, as the sequences of four Serbian AGs were clustered with their respective representatives and the resulting topology was consistent with previously published [13,44]. Phylogenetic analyses of BNR diversity have been extensively studied [31,44]. Sharon et al. [12] conducted the most comprehensive study of 21 AGs, which served as an important reference for global and our BNR research.

Despite the fact that AG-A was the most frequently detected BNR in Serbia (10 out of 16 locations), only strawberry infection was documented. Compared to previous studies [20], AG-A was detected at six new locations with a significant disease incidence (up to 35%), which correlates with its increasing importance for strawberries, a fact also observed in California, South Africa, Italy, and Israel, with a detection frequency of up to 70% [31,54,55]. The host range of AG-A isolates is mainly associated with the strawberry, but in other parts of the world, it has also been described as a pathogen in sugar beet, beans, peas, sunflowers, and other crops [2,22].

A BNR AG-A haplotype analysis was performed due to its high prevalence and wide distribution in Serbia. Our study revealed a high diversity in the population of BNR AG-A and the well-established 24 separate haplotypes worldwide. The Serbian population of BNR AG-A is also diverse, as our isolates were clustered into nine haplotypes. Published data on the morphological and phylogenetic diversity of BNR AG-A show the variability of the population. Li et al. [50] reported the morphological diversity of BNR AG-A and described three morphotypes based on colony morphology. Phylogenetic analyses by Manici and Bonora [31] confirmed the intra-AG variability of Italian AG-A isolates and showed that they are divided into four sub-clusters. Vojvodić et al. [20] confirmed that isolates from Serbia also show genetic variability and are divided into three defined sub-clusters. To date, there are no data on the haplotype composition of BNR AG-A worldwide, and our analyses represent the first attempt to characterize its status. Regarding the diversity of BNR AGs, Li et al. [28] published haplotype analyses of BNR AG-D and indicated the existence of three evolutionary origins. Ceresini et al. [25] reported two sister populations of *R. solani* AG-3 from potato and tobacco, representing two genetically distinct and historically divergent lineages that evolved depending on their Solanaceaeous hosts. Wang et al. [27] and Wei et al. [26] reported 43 and 12 haplotypes, respectively, in the Chinese population of MNR AG-1 IC from rice.

BNR AG-G has a broad host range that includes several vegetable species [2], such as the tomato and bean [19,56], which is comparable to the situation in Serbia as it was detected at three locations and on three different host plants: the tomato, bean, and cherry. AG-G on cherries caused chlorosis and necrotic zones on the leaves, as well as necrosis on the stem base, roots, and root hairs. Among the host plants from the Rosaceae family, AG-G has been shown to be pathogenic in strawberries [22] and apples [22,45]. To date, there is little evidence of *Rhizoctonia* spp. being pathogenic to *Prunus* spp. There is only one report of an *Rhizoctonia* spp. of unknown AG as an endophyte on cherries [57], and MNR AG-1-IC causing water-soaked lesions on the roots of *Prunus amygdalus × Prunus persica* [58] in Tunisia, followed by the subsequent collapse of the entire plant. Our results on AG-G as a cherry pathogen could be particularly valuable with regard to a possible endophytic lifestyle and a possible latent infection that spreads through the planting material and later transforms into a pathogen, especially in stressed or damaged plants.

Similar to Serbia, BNR AG-F has already been detected on peppers in Turkey, causing drying and rotting of the plants [56], but has also been detected on beans [19] and tobacco [59], indicating a preference for vegetable crops. These findings suggest that AG-F could pose a considerable risk to vegetable crops, especially in greenhouses.

Very little information is available on BNR AG-U, which has only been associated with a few hosts worldwide. So far, BNR AG-U has been detected on ornamental plants in Japan and in the southern part of the USA [53,60,61], on onions and carrots in Japan [61,62], on potatoes in China [63], and as a cause of apple root rot in Italy [45]. Our report on meadow grass is the first report on any monocotyledonous host. This study also provides the first detailed insight into the pathogenic properties and experimental host range of BNR AG-U.

Based on our host range tests, significant differences in the specificity and aggressiveness of *Rhizoctonia* spp. isolates within and between different AGs were observed. The differences have been previously documented either based on seedling inoculation [64,65,66] or derived from natural infections and predicted host ranges [1,3,10,67]. Although most BNRs were originally described as pathogens of monocotyledonous hosts [16,68], the host range of the four BNR AGs characterized in our study showed that the isolates exhibited lower aggressiveness on wheat and maize compared to the dicotyledonous hosts tested. Of the four isolates of AG-A, none were able to infect cabbage seedlings, and at least one isolate was unable to infect tobacco, lettuce, peas, and maize, but were most aggressive on sunflowers and sugar beet, and partially on bean seedlings. The three tested AG-G isolates were the most aggressive on the majority of the test plants, with slight differences between them, while AG-F and AG-U showed medium to high aggressiveness on all 14 host plants. The tested plants from the Solanaceae, Brassicaceae, and Poaceae families showed similar response rates under the same AGs, while the plants from the Asteraceae and Fabaceae families responded differently. Bean seedlings exhibited high susceptibility to all AGs tested. Similarly, bean seedlings were highly susceptible to AG-A, AG-F, and AG-G in the artificial inoculations conducted in Brazil [24], and were found to be very susceptible hosts for different *Rhizoctonia* AGs: AG-1 IB, AG-2-1, AG-2-2, AG-5, AG-A, AG-G, AG-E, AG-I, and AG-K [10,56]. The results obtained with artificial inoculations correspond to the results on plant susceptibility to natural infections. In natural BNR infections, the most susceptible hosts are beans, peas, tomatoes, peppers, wheat, and maize [19,68,69,70,71,72]. In the study by Blanco et al. [24], the maize seedlings showed similar susceptibility to BNR AG-A, AG-F, and AG-G as our isolates.

The BNR AGs tested showed varying degrees of aggressiveness on leaf petioles, allowing comparisons between isolates within AGs. In general, all four BNR AGs were found to be the most aggressive against strawberry petioles, but also against tomatoes and sunflowers. The least aggressive isolates belonged to BNR AG-A. The degree of aggressiveness in this experiment was similar to the experiment on young seedlings. When comparing the two methods of assessing pathogenicity and aggressiveness, the results were consistent, except for beans, where isolates were more aggressive towards seedlings than petioles. Aggressiveness testing on different petioles proved to be a fast and convenient method compared to the seedling method, providing accurate results in a short time, which is particularly suitable for rapid pathogenicity testing on a large number of isolates.

Soil-borne pathogens are often more difficult to manage than airborne ones as they are usually discovered when severe damage has already occurred [73]. Their frequent polyphagous nature and long survival time in plant residues or in the soil, as well as the non-specific symptoms that they cause, make management even more demanding. For many soil-borne pathogens, growing resistant varieties, when available, is an effective management option. So far, there are only a few or no varieties resistant to *Rhizoctonia* spp. [74]. To prevent soil-borne disease outbreaks, crop rotation is recommended as a valuable practice. However, for appropriate crop sequence selection, identifying which AGs are present in the soil and which crops/varieties they are pathogenic to is of crucial importance. Chemical control could also be an effective method of controlling some soil-borne diseases, provided that fungicide selection is based on the precise identification of the causal AG, as it was shown that different AGs respond differently to the available fungicides [3].

Besides the first detection of three AGs in Serbia, AG-F on pepper, AG-G on beans and tomatoes, and AG-U on apples, this study reports the newly discovered hosts of two AGs. For the first time in the world, cherries and meadow grass were found as hosts of AG-G and AG-U, respectively. In addition, differences in the specificity and aggressiveness of *Rhizoctonia* spp. isolates within and between different AGs were documented. New hosts and observed aggressiveness towards a much larger number of plant species than previously recognized suggest that the impact of BNR on different hosts in the world and in Serbia may be underestimated. To improve our options and protocols for effective disease management, more knowledge concerning AG diversity, host range, and aggressiveness is needed.

## Figures and Tables

**Figure 1 jof-11-00410-f001:**
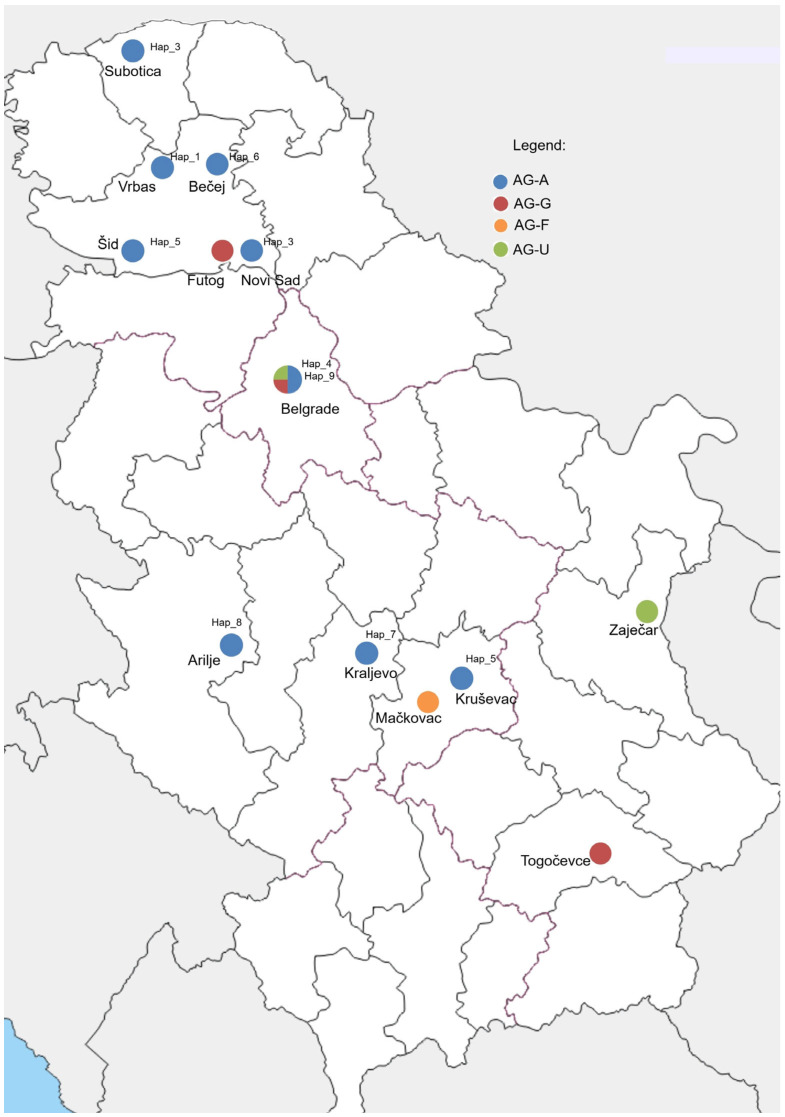
Geographic distribution of localities in Serbia included in the survey and detected binucleate Rhizoctonia AGs and AG-A haplotypes.

**Figure 2 jof-11-00410-f002:**
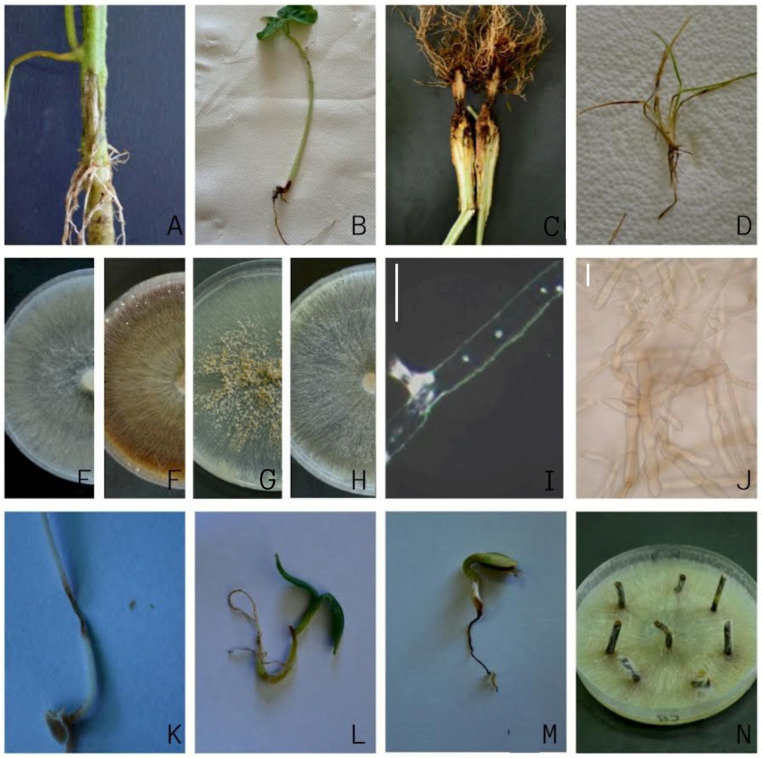
Binucleate *Rhizoctonia* AGs: (**A**–**D**) root and root neck necrosis of the tomato, bean, pepper, and *Poa annua*, respectively; (**E**–**H**) seven-day-old colonies on PDA of *Rhizoctonia* spp. AG-A, AG-G, AG-F, and AG-U, respectively; (**I**) nuclei in hypha (bar = 20 µm); (**J**) moniloid cells in 5-day-old colonies of *Rhizoctonia* spp. AG-G (bar = 20 µm); (**K**,**L**) root necrosis in tomato plants inoculated with *Rhizoctonia* spp. AG-G and AG-F, respectively; (**M**) root necrosis of a sunflower inoculated with *Rhizoctonia* spp. AG-G; (**N**) pathogenicity testing on the detached leaf petioles of strawberry *Rhizotonia* AG-A.

**Figure 3 jof-11-00410-f003:**
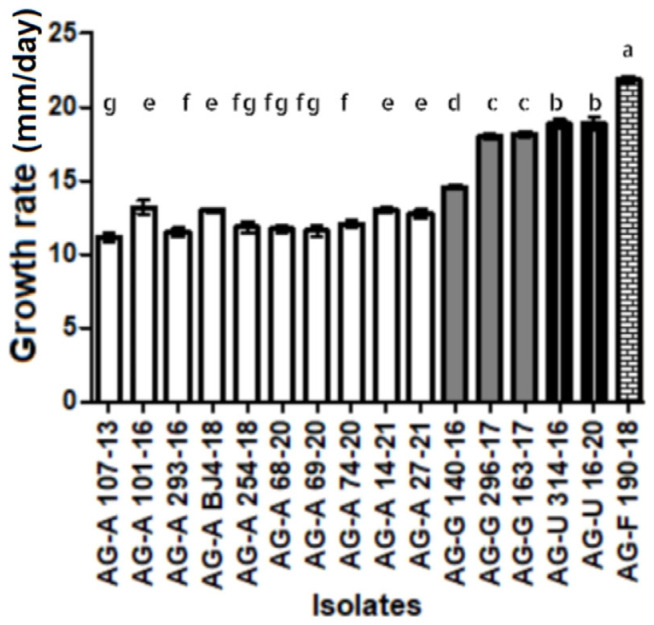
Growth rate of isolates of binucleate *Rhizoctonia* spp. AG-A, AG-G, AG-F, and AG-F on PDA at 24 °C in the dark. Bars represent standard deviation. Values marked with the same letter do not differ significantly.

**Figure 4 jof-11-00410-f004:**
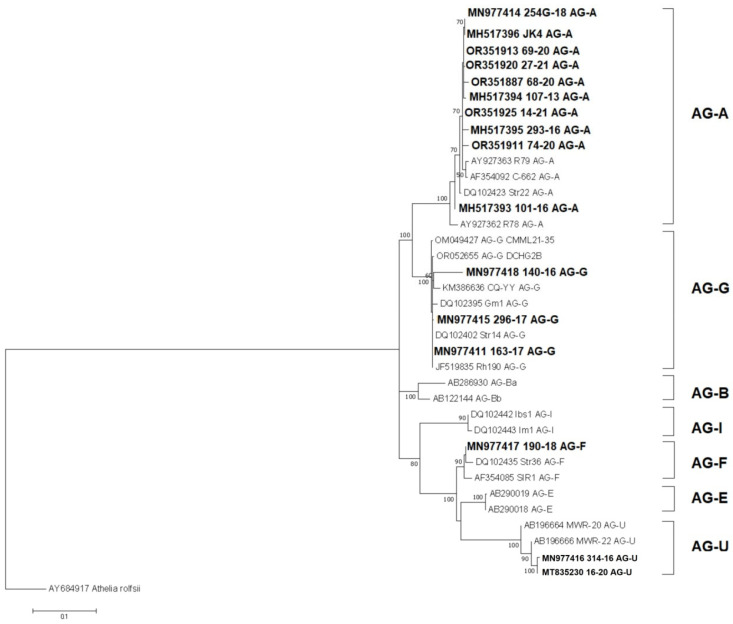
Maximum likelihood phylogenetic tree of the internal transcribed spacer rDNA sequences of 16 Serbian and 21 reference isolates of binucleate *Rhizoctonia* spp. from nine AGs, and the outgroup taxa *Athelia rolfsii.* The tree was generated in MEGA X using Kimura’s two-parameter model. Bootstrap analyses were performed with 1000 replicates, and bootstrap values (>50%) are shown next to the corresponding branches. Serbian isolates are in bold.

**Figure 5 jof-11-00410-f005:**
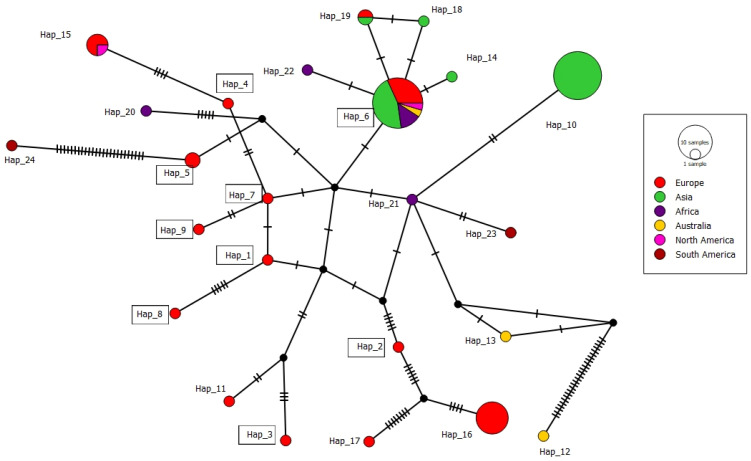
Median-joining network showing the phylogenetic relationships between the haplotypes of isolates of *Rhizoctonia* spp. AG-A isolates available in GenBank. The black nodes represent median vectors (missing or not sampled haplotypes) required to connect existing haplotypes within the network with maximum parsimony. Details of the sequences and haplotypes are shown in Table 3. Only the numbers of significant haplotype groups are listed. Haplotypes to which Serbian isolates belong are labeled with rectangles. Different colors indicate the proportions within the haplotypes from different continents.

**Figure 6 jof-11-00410-f006:**
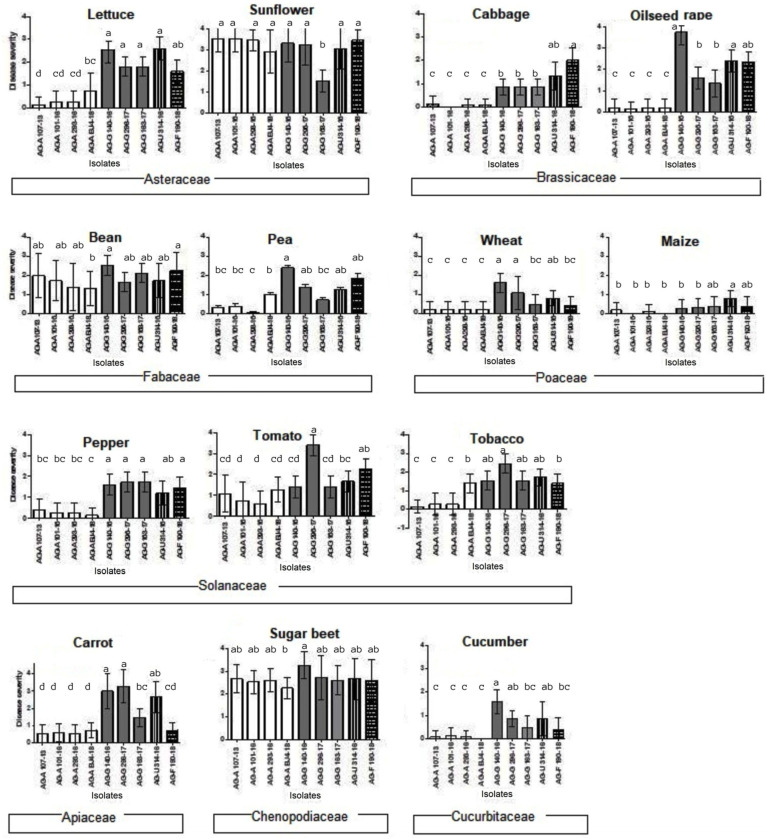
Root rot disease severity on different host plant seedlings caused by four binucleate *Rhizoctonia* AGs (AG-A, AG-G, AG-F, and AG-U) isolates rated following the scale: 0—no reaction; 1—up to 30% of roots affected; 2—up to 40% of roots affected; 3—total of 40–60% of roots affected; and 4—roots and entire plantlet necrotic and decayed. Bars represent standard deviation. Values marked with the same letter do not differ significantly.

**Figure 7 jof-11-00410-f007:**
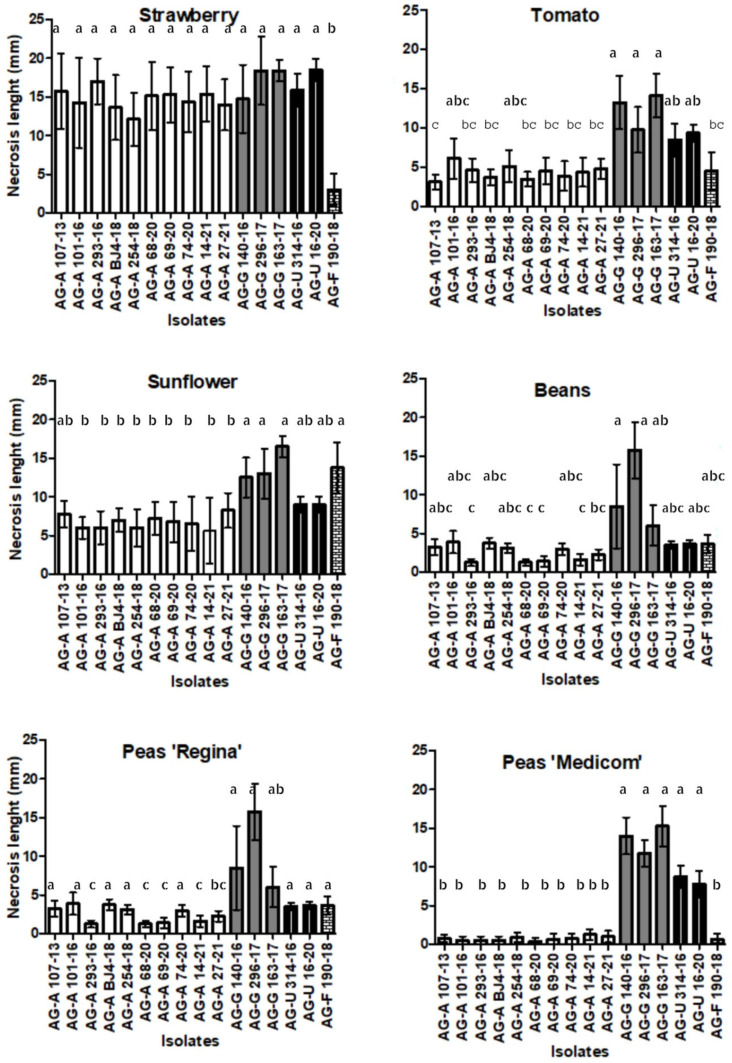
Aggressiveness of four binucleate *Rhizoctonia* AGs (AG-A, AG-G, AG-F, and AG-U) on detached leaf petioles of strawberries, sunflowers, tomatoes, beans, and peas (necrosis length mm/day). Bars represent standard deviation. Values marked with the same letter do not differ significantly.

**Table 1 jof-11-00410-t001:** Sample data collected in Serbia and disease intensity.

Year	Locality	District	Host	Disease Incidence (%) **	Number of Collected Samples	Number of Isolates	AG ***
2013	Subotica	North Bačka	Strawberry	20	15	12	AG-A
2016	Belgrade 1	Belgrade	Strawberry	30	13	9	AG-A
	Togočevce	Jablanica	Tomato	25	6	4	AG-G
	Novi Sad	South Bačka	Strawberry	30	12	9	AG-A
	Zaječar	Zaječar	Apple	5	7	3	AG-U
2017	Belgrade 2	Belgrade	Cherry	5	7	5	AG-G
	Futog 2	South Bačka	Bean	15	8	5	AG-G
2018	Kruševac	Rasina	Strawberry	25	13	10	AG-A
	Mačkovac	Rasina	Papper	15	7	4	AG-F
	Šid	Sremski	Strawberry	35	10	8	AG-A
2020	Belgrade 3	Belgrade	Meadow grass	30	14	6	AG-U
	Belgrade 4	Belgrade	Strawberry	25	12	10	AG-A
	Kraljevo	Raška	Strawberry	20	10	7	AG-A
	Arilje	Zlatibor	Strawberry	25	11	9	AG-A
2021	Vrbas	South Bačka	Soil (strawberry *)	-	10	5	AG-A
	Bečej	South Bačka	Soil (strawberry)	-	8	6	AG-A

* Previous crop; ** average disease incidence estimated by walking through the crop in a zigzag pattern and randomly rating 100 plants in three replicates; *** identified by ITS sequencing.

**Table 2 jof-11-00410-t002:** Binucleate *Rhizoctonia* spp. and related internal transcribed spacer sequences derived from GenBank and included in the phylogenetic analysis.

Isolate	*Rhizoctonia* spp. AG	Host	Country	Acc. No. GenBank	Literature Reference
R78	AG-A	Strawberry	Italy	AY927362	[31]
R79	AG-A	Strawberry	Italy	AY927363	[31]
C-662	AG-A	Soil	Japan	AF354092	[31]
Str22	AG-A	Strawberry	Israel	DQ102423	[12]
107-13	AG-A	Strawberry	Serbia	MN517394	[20]
101-16	AG-A	Strawberry	Serbia	MH517393	[20]
293-16	AG-A	Strawberry	Serbia	MH517395	[20]
BJ4-18	AG-A	Strawberry	Serbia	MH517396	[20]
254-18	AG-A	Strawberry	Serbia	MN977414	This study
68-20	AG-A	Strawberry	Serbia	OR351887	This study
69-20	AG-A	Strawberry	Serbia	OR351913	This study
74-20	AG-A	Strawberry	Serbia	OR351911	This study
14-21	AG-A	Strawberry	Serbia	OR351925	This study
27-21	AG-A	Strawberry	Serbia	OR351920	This study
Scl2	AG-B	Rice	Japan	AB286930	[12]
C-350	AG-B	Rice	Japan	AB122144	[12]
Gm1	AG-G	Strawberry	USA	DQ102395	[12]
Str14	AG-G	Strawberry	Israel	DQ102402	[12]
DCHG2B	AG-G	Grapevine	USA	OR052655	[42]
CMML21-35	AG-G	Japanese bay tree	South Korea	OM049427	[43]
CQ-YY	AG-G	Potato	China	KM386636	[44]
Rh190	AG-G	Apple rootstock	Italy	JF519832	[45]
140-16	AG-G	Tomato	Serbia	MN977418	This study
163-17	AG-G	Cherry	Serbia	MN977411	This study
296-17	AG-G	Bean	Serbia	MN977415	This study
Oc-1	AG-E	Wood sorrels	Japan	AB290019	[12]
Lu-5	AG-E	Flax	Japan	AB290018	[12]
Str36	AG-F	Strawberry	Israel	DQ102435	[12]
SIR	AG-F	Sweet potato	Japan	AF354085	[12]
190-18	AG-F	Papper	Serbia	MN977417	This study
MWR-22	AG-U	Rose	Japan	AB196666	[12]
MWR-20	AG-U	Rose	Japan	AB196664	[12]
314-16	AG-U	Apple	Serbia	MN977416	This study
16-20	AG-U	Meadow grass	Serbia	MT835230	This study
Ibs1	AG-I	Soil	Israel	DQ102442	[12]
Im1	AG-I	Strawberry	USA	DQ102443	[12]
FSR-052	*Athelia rolfsi*	Lily	Taiwan	AY684917	[12]

**Table 3 jof-11-00410-t003:** Binucleate *Rhizoctonia* spp. AG-A sequences used in haplotype analyses.

Haplotype	Accession Number	Continent	Country	Host
Hap_1	OR351925	Europe	Serbia	strawberry
Hap_2	MH517393	Europe	Serbia	strawberry
Hap_3	MH517395	Europe	Serbia	strawberry
Hap_4	MH517394	Europe	Serbia	strawberry
Hap_5	MH517396	Europe	Serbia	strawberry
Hap_5	MN977414	Europe	Serbia	strawberry
Hap_6	OR351920	Europe	Serbia	strawberry
Hap_6	OL840587	Europe	France	Pea
Hap_6	JQ859848	Australia	Australia	strawberry
Hap_6	FR734303	Asia	Turkey	Tobacco
Hap_6	FR734301	Asia	Turkey	Tobacco
Hap_6	FR734300	Asia	Turkey	Tobacco
Hap_6	FR734299	Asia	Turkey	Tobacco
Hap_6	FR734293	Asia	Turkey	Tobacco
Hap_6	FR734288	Asia	Turkey	Tobacco
Hap_6	AY927349	Europe	Italy	strawberry
Hap_6	AY927342	Europe	Italy	strawberry
Hap_6	AY927330	Europe	Italy	strawberry
Hap_6	AY927328	Europe	Italy	strawberry
Hap_6	AY927326	Europe	Italy	strawberry
Hap_6	AY927322	Europe	Italy	strawberry
Hap_6	JX073669	Asia	China	sugar beet
Hap_6	JX073668	Asia	China	sugar beet
Hap_6	AB196663	Asia	Japan	Rose
Hap_6	KJ777575	Africa	South Africa	Potato
Hap_6	KJ777644	Africa	South Africa	Potato
Hap_6	KJ777636	Africa	South Africa	Potato
Hap_6	OL471747	North America	USA	Potato
Hap_7	OR351913	Europe	Serbia	strawberry
Hap_8	OR351911	Europe	Serbia	strawberry
Hap_9	OR351887	Europe	Serbia	strawberry
Hap_10	PP902547	Asia	Turkey	strawberry
Hap_10	PP902546	Asia	Turkey	strawberry
Hap_10	PP902545	Asia	Turkey	strawberry
Hap_10	PP902544	Asia	Turkey	strawberry
Hap_10	PP902543	Asia	Turkey	strawberry
Hap_10	PP902542	Asia	Turkey	strawberry
Hap_10	PP902541	Asia	Turkey	strawberry
Hap_10	PP902540	Asia	Turkey	strawberry
Hap_10	PP902539	Asia	Turkey	strawberry
Hap_10	PP902538	Asia	Turkey	strawberry
Hap_10	PP902537	Asia	Turkey	strawberry
Hap_10	PP902536	Asia	Turkey	strawberry
Hap_10	PP902535	Asia	Turkey	strawberry
Hap_10	PP902534	Asia	Turkey	strawberry
Hap_10	PP902533	Asia	Turkey	strawberry
Hap_10	PP902532	Asia	Turkey	strawberry
Hap_10	PP902531	Asia	Turkey	strawberry
Hap_10	PP902530	Asia	Turkey	strawberry
Hap_10	PP902529	Asia	Turkey	strawberry
Hap_10	PP902528	Asia	Turkey	strawberry
Hap_11	OR231136	Europe	Albania	strawberry
Hap_12	JQ859850	Australia	Australia	strawberry
Hap_13	JQ859849	Australia	Australia	strawberry
Hap_14	FR734298	Asia	Turkey	Tobacco
Hap_15	AY927363	Europe	Italy	strawberry
Hap_15	AY927360	Europe	Italy	strawberry
Hap_15	AY738628	Europe	Italy	strawberry
Hap_15	OL471751	North America	USA	Potato
Hap_16	AY927362	Europe	Italy	strawberry
Hap_16	AY927361	Europe	Italy	strawberry
Hap_16	AY927347	Europe	Italy	strawberry
Hap_16	AY927343	Europe	Italy	strawberry
Hap_16	AY927339	Europe	Italy	strawberry
Hap_16	AY927338	Europe	Italy	strawberry
Hap_16	AY927337	Europe	Italy	strawberry
Hap_16	AY927335	Europe	Italy	strawberry
Hap_16	AY927315	Europe	Italy	strawberry
Hap_17	AY927358			
Hap_18	KP893156	Asia	China	strawberry
Hap_19	AB196661	Europe	Italy	hop bush
Hap_20	KJ777564	Africa	South Africa	Potato
Hap_21	KJ777631	Africa	South Africa	Potato
Hap_22	KJ777641	Africa	South Africa	Potato
Hap_23	KM065554	South America	Brazil	Soil
Hap_24	KM065539	South America	Brazil	Soil

## Data Availability

Dataset available on request from the authors.

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
