# Peer review of "Diversity of Binucleate Rhizoctonia spp. and Population Structure of AG-A in Serbia"

_jof, 2025, doi:10.3390/jof11060410_

Round 1
Reviewer 1 Report
Manuscript entitled “Diversity of binucleate Rhizoctonia spp. and population structure of AG-A in Serbia”. This study isolated different binucleate Rhizoctonia spp. strains from different plants and soil, and investigate the diversity and population structure of AG-A in Serbia. Major points need to be addressed before it can be accepted.
- Line 86. Plant samples were collected from 2013-2021. Were Rhizoctonia spp. strains isolated at the sampling year or all at 2021?
- Line 99. When collected soil samples, are there any strawberry plants growth?
- Table 1. The method to investigate disease incidence should be provided.
- Table 1. The criterion to distinguish different types of AGs should be provided.
- Line119. This study isolated 112 Rhizoctonia spp. strains, why choose these 10 isolates for pathogenicity test?
- Line121. Reference 31. The modified method should be briefly introduced.
- Line 149-151. The sequences of each prime pairs should be provided.
- 2.6. This study selected 9 isolates for seedling inoculation and 16 isolates for detached leaf inoculation. Are there any strains repeated for seedling inoculation and detached leaf inoculation? Why not use 16 isolates for seedling inoculation?
- Line 229. Italic for p<0.05.
- Figure 2. The author should add the meaning of different characters in figure legends. Moreover, the position of these characters should closely under the bar of chart.
- Line285-286. Why the amplification of LSU, RPB2, tef-1α and atp6 was not possible for isolates from all four AGs?
Manuscript entitled “Diversity of binucleate Rhizoctonia spp. and population structure of AG-A in Serbia”. This study isolated different binucleate Rhizoctonia spp. strains from different plants and soil, and investigate the diversity and population structure of AG-A in Serbia. Major points need to be addressed before it can be accepted.
- Line 86. Plant samples were collected from 2013-2021. Were Rhizoctonia spp. strains isolated at the sampling year or all at 2021?
- Line 99. When collected soil samples, are there any strawberry plants growth?
- Table 1. The method to investigate disease incidence should be provided.
- Table 1. The criterion to distinguish different types of AGs should be provided.
- Line119. This study isolated 112 Rhizoctonia spp. strains, why choose these 10 isolates for pathogenicity test?
- Line121. Reference 31. The modified method should be briefly introduced.
- Line 149-151. The sequences of each prime pairs should be provided.
- 2.6. This study selected 9 isolates for seedling inoculation and 16 isolates for detached leaf inoculation. Are there any strains repeated for seedling inoculation and detached leaf inoculation? Why not use 16 isolates for seedling inoculation?
- Line 229. Italic for p<0.05.
- Figure 2. The author should add the meaning of different characters in figure legends. Moreover, the position of these characters should closely under the bar of chart.
- Line285-286. Why the amplification of LSU, RPB2, tef-1α and atp6 was not possible for isolates from all four AGs?
Author Response
-
- Line 86. Plant samples were collected from 2013-2021. Were Rhizoctonia spp. strains isolated at the sampling year or all at 2021?
Answer: Samples were analyzed following the sampling and additional explanation was included in line 93.
- Line 99. When collected soil samples, are there any strawberry plants growth?
Answer: The soil samples were collected from fields where strawberries had previously been grown and where no strawberry plants were present.
- Table 1. The method to investigate disease incidence should be provided.
Answer: Accepted and added to the table legend.
- Table 1. The criterion to distinguish different types of AGs should be provided.
Answer: Accepted and added to the table legend.
- Line119. This study isolated 112 Rhizoctonia spp. strains, why choose these 10 isolates for pathogenicity test?
Answer: Only isolates belonging to the same AG and having a uniform morphology were obtained from each locality, so that a representative isolate from each host plant/field combination was selected for further studies.
- Reference 31. The modified method should be briefly introduced.
Answer: Since the modification refers to the quantity of the inoculum, the word “modified” in line 122 is deleted.
- Line 149-151. The sequences of each prime pairs should be provided.
Answer: Accepted and added in lines 153-159.
- 2.6. This study selected 9 isolates for seedling inoculation and 16 isolates for detached leaf inoculation. Are there any strains repeated for seedling inoculation and detached leaf inoculation? Why not use 16 isolates for seedling inoculation?
Answer: All 9 isolates used in the inoculation of seedlings are also used for the inoculation of detached leaf petioles. The first test was used to test the experimental host range and virulence, while the second test was used to check reliability of the method, as it is much easier and faster for large scale testing.
- Line 229. Italic for p<0.05.
Answer: Accepted and changed.
- Figure 2. The author should add the meaning of different characters in figure legends. Moreover, the position of these characters should closely under the bar of chart.
Answer: Accepted and explanation added in lines 302-303.
- Line285-286. Why the amplification of LSU, RPB2, tef-1α and atp6 was not possible for isolates from all four AGs?
Answer: Difficulties in the amplification of various Rhizoctonia AGs have already been noted and described by other researchers (Gonzalez, D.; Rodriguez-Carres, M.; Boekhout, T.; Stalpers, J.; Kuramae, E.E.; Nakatani, A.K.; Vilgalys, R.; Cubeta, M.A.: Phylogenetic relationships of Rhizoctonia fungi within the Cantharellales. Fungal Biol. 2016, 120, 603-619. DOI: https://doi.org/10.1016/j.funbio.2016.01.012).
Reviewer 2 Report
The manuscript describes the isolation and characterization of Rhizoctonia solani strains from different locations in Serbia. Pathogenicity and virulence tests of the isolates obtained were performed in different plant species. The haplotypes of the isolates were identified by analyzing the ITS region, and a phylogenetic analysis was performed on the same genetic region.
Among the manuscript's contributions, the authors mention the first-time description in Serbia of AG-F on pepper, AG-G on bean and tomato, and AG-U on apple. They also mention the first-time description of cherry and meadow grass as hosts of AG-G and AG-U, respectively.
The submitted manuscript is suitable for its publication in the Journal of Fungi after a minor review. Below are some specific comments for the authors' consideration.
- I suggest to the authors include a three-level map (continental, country, sampling locations ubication) showing the relative positions of the sampling locations. This map is desirable in this kind study because it can help to visualize dispersal patterns or geographical restrictions of the genotypes found. In relation to this comment, can the results show any possible pattern of dispersion of the haplotypes found between the sampling sites?
- Please add a scale on all photographs in Figure 1 that includes plant structures or photomicrographs of mycelium.
- Although it is common notation to indicate statistically significant differences, please indicate in the figure caption the meaning of the letters in each bar in figures 2, 5 and 6.
- Please write the species name Athelia rolfsi correctly in the figure caption of figure 3, it should say Athelia rolfsii
Author Response
- I suggest to the authors include a three-level map (continental, country, sampling locations ubication) showing the relative positions of the sampling locations. This map is desirable in this kind study because it can help to visualize dispersal patterns or geographical restrictions of the genotypes found. In relation to this comment, can the results show any possible pattern of dispersion of the haplotypes found between the sampling sites?
Answer: Accepted and added as Figure 1. The additional explanations are included in the lines 244 and 343. Accordingly, the names of all other figures are changed as well as referrals in the manuscript.
- Please add a scale on all photographs in Figure 1 that includes plant structures or photomicrographs of mycelium.
Answer: Accepted and changed
- Although it is common notation to indicate statistically significant differences, please indicate in the figure caption the meaning of the letters in each bar in figures 2, 5 and 6.
Answer: Accepted and included in lines 302-303, 407-408 and 416.
- Please write the species name Athelia rolfsi correctly in the figure caption of figure 3, it should say Athelia rolfsii
Answer: Accepted and changed.
Round 2
Reviewer 1 Report
Accept in present form
Accept in present form